# Factors influencing the performance of cardiopulmonary resuscitation by lay rescuers: A qualitative study based on the Theory of Planned Behavior

Zhikang Zhong[1], Weiwei Li[2], Lin Zhou[1], Liang Liu [1]*

**1** Department of Nephrology, The Second Xiangya Hospital of Central South University, Changsha, Hunan, China, **2** Department of Geriatrics, People's Hospital of Wuhan University, Wuhan, Hubei, China

* 507242@csu.edu.cn

## Abstract

### Aims

This study investigated the underlying causes for the current low rate of cardiopulmonary resuscitation (CPR) implementation among lay rescuers. Specifically, the study aimed to identify the factors that influence both the intention to perform CPR and the subsequent translation of this intention into actual behavior. The ultimate goal was to propose strategies to significantly enhance the rate of bystander CPR.

### Methods

A qualitative study employing purposive sampling was undertaken, with semi-structured interviews conducted between January 1, 2022, and May 20, 2022. The collected data were analyzed thematically using Nvivo software.

### Results

A total of 29 lay rescuers participated in this study, and 11 thematic categories were identified across the four constructs of the Theory of Planned Behavior (TPB). Within the domain of attitudes, two primary themes emerged, each comprising two subthemes: (1) Emotional resistance to the death event and (2) recognition of the potential benefits of rescue, including the prospect of external rewards and the realization of intrinsic self-worth. For perceived norms, three themes were identified: (1) The bystander effect, (2) the encouraging influence of peers, and (3) personal motivation arising from one's career. With respect to perceived behavioral control (PBC), three themes and two subthemes emerged: (1) Consideration of potential consequences, encompassing concerns about legal disputes and the fear of disease transmission; (2) the dual impact of public opinion pressure; (3) low self-efficacy. Finally, in relation to actual control, three themes and two subthemes were demonstrated: (1)

**Data availability statement:** All additional Supporting information files are accessible through the openICPSR database at the following link: https://doi.org/10.3886/E213184V2.

**Funding:** This study was supported by a scientific research project from the Hunan Provincial Health Commission (Project No. D202303056656), which was awarded the project to Ms Zhou, which provided financial support for the subsequent APC.Lin Zhou, as the principal investigator, provided significant contributions to the formal analysis of the data, the interpretation of results, and the revision of the manuscript.

**Competing interests:** The authors have declared that no competing interests exist.

Acquisition of CPR skills; (2) the influence of situational factors at the scene, including both supportive elements such as public facilities or procedures and obstructive factors, such as a diminished sense of responsibility at specific locations; (3) the impact of the patient's condition.

## Conclusions

This study extends the TPB in the context of bystander CPR interventions by (1) integrating both emotional (death anxiety) and cognitive dimensions of attitude formation under stress, (2) reconceptualizing PBC as context-dependent, influenced by legal risks and automated external defibrillator (AED) accessibility, and (3) addressing the tension between the bystander effect and moral obligation through normative adjustments. To improve intervention rates, it is crucial to target emotional barriers (virtual reality (VR)-based anxiety reduction) and address systemic constraints (AED availability and simplified Good Samaritan laws) while leveraging artificial intelligence tools to reinforce positive norms. Future work is encouraged to validate the model's cross-cultural applicability and assess interventions such as community AED programs and VR training aimed at bridging the gap between bystander hesitation and timely CPR delivery.

## 1. Introduction

Out-of-hospital cardiac arrest (OHCA) is a leading global cause of death [1]. According to *the 2020 American Heart Association Guidelines for Cardiopulmonary Resuscitation and Emergency Cardiovascular Care* (*AHA Guidelines*), an estimated 350,000 adults in the United States experienced OHCA in 2015 [2]. In China, emergency medical services responded to approximately 97.1 OHCA cases per 10 million people in 2020, with a mere 1.2% survival rate and a 0.8% rate of good neurological prognosis [3]. Early cardiopulmonary resuscitation (CPR) significantly enhances the survival chances of patients with OHCA, doubling their likelihood of survival [4]. Although the optimal response time for resuscitation is within 4 min of cardiac arrest, the median response time of emergency medical services in mainland China is 12 min [3]. Consequently, enhancing the CPR implementation rate among lay rescuers is of critical importance. However, in China, this rate remains relatively low, ranging from 2.8% to 11.4% [5], far lower than that of other developed countries [6,7].

The primary strategies to improve CPR delivery rates involve promoting and providing CPR skills training. However, existing training programs, which primarily focus on knowledge and skill transfer, may ineffectively boost CPR implementation intentions. As demonstrated by Robert Swor and colleagues, participants who undergo CPR training frequently face challenges in translating their skills to real-world scenarios, leading to inadequate CPR implementation [8]. Moreover, despite large-scale CPR education campaigns across the United States, only 30%–40% of patients with OHCA ultimately receive CPR [9]. Recognizing this challenge, the *AHA Guidelines*

emphasize the need to improve CPR intentions [10]. Intention, defined as an individual's perceived likelihood of performing a specific task with a specific intent, is crucial in promoting behavior. For instance, the *AHA Guidelines* recommend chest-only CPR [11] to mitigate the risk of disease transmission during mouth-to-mouth resuscitation, thereby enhancing lay rescuers' willingness to perform and ultimately boosting CPR implementation rates [12]. Although positive intentions are closely associated with behavior, numerous barriers can hinder the translation of these intentions into action. Consequently, individuals with strong intentions may still fail to execute CPR in real-world situations [9]. To effectively improve public CPR implementation rates, it is crucial to identify the factors influencing lay rescuers' intentions and to examine the obstacles that impede the translation of these intentions into actual behavior.

The Theory of Planned Behavior (TPB), proposed by Ajzen in 1985 [13], offers a framework for understanding the complex relationship between human intentions and actual behavior [14,15]. It is widely applied across various fields, including healthcare behaviors such as hand hygiene and diabetes management [16–18]. However, notably, relatively few studies have examined the experiences of lay rescuers performing CPR within the framework of this theory. TPB posits that individual behavior is influenced not only by personal attitudes and experiences but also by perceived social pressures. The model effectively predicts factors influencing an individual's behavioral intentions in health-related contexts [19]. However, while the intention is a precursor to behavior, it is not a definitive predictor. Positive intentions do not guarantee behavior, as intervening factors can affect the conversion of intention into action. As illustrated in Fig 1, the realization of a specific behavior depends on the strength of an individual's intention, which is shaped by various factors, including knowledge, beliefs, perceived consequences, self-control, and real-world obstacles. By identifying factors that either enhance or diminish lay rescuers' intention to perform CPR, we can effectively improve public intentions and the subsequent conversion of these intentions into behavior.

While the TPB effectively accounts for bystanders' translation of CPR intentions into actual behavior through attitudes, subjective norms, and perceived behavioral control (PBC), it may inadequately address the psychological processes—such as emotion-cognition interactions—that are activated during real-world OHCA scenarios. Specifically, OHCA events can provoke acute emotional barriers (anxiety and fear) that may disrupt rational decision-making [20]. Additionally, contextual risk evaluations, including factors such as bystander effect and legal concerns, must be comprehensively

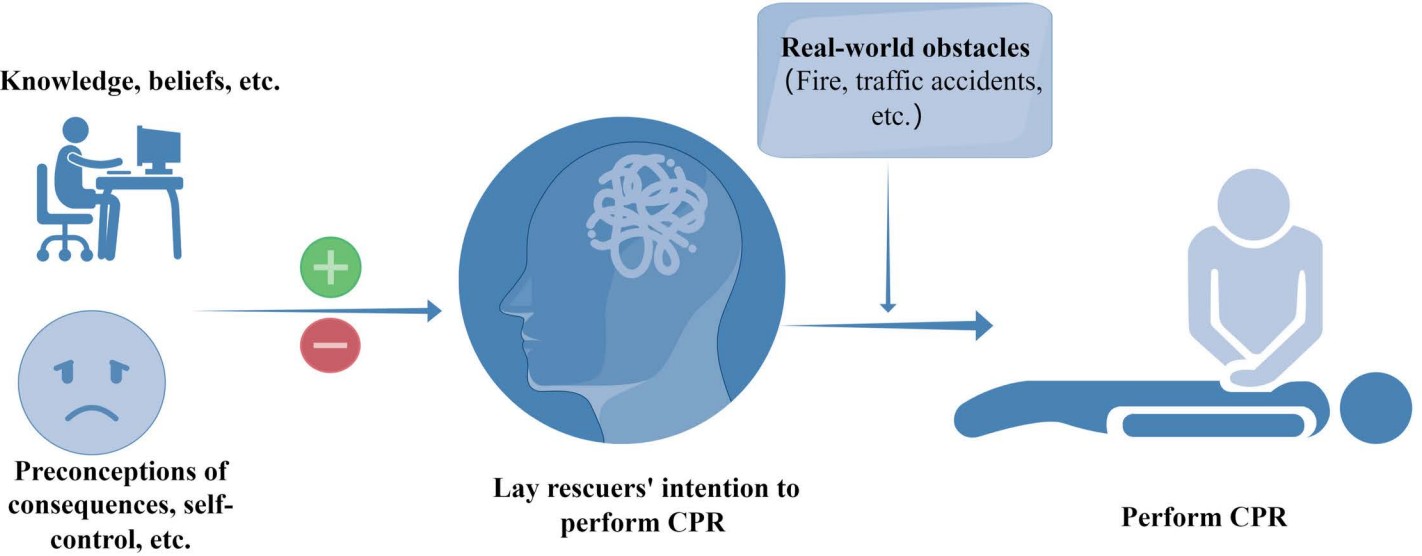

**Fig 1. Bystander CPR Intention-Behavior Model (generated by Figdraw).**

examined to elucidate how intention-behavior transitions are modulated in practice [21]. Investigating the roles of these factors in shaping emotional responses and mediating situational dynamics may advance TPB's applicability and contribute to effective bystander CPR implementation.

Therefore, this study employed semi-structured interviews and a TPB-based approach to explore the risk and protective factors influencing lay rescuers' intention to perform CPR. Furthermore, we examined the interplay between emotions and cognition, along with the influence of real-world situational variables on bystander CPR performance. By thoroughly understanding these factors, clinical and training organizations can develop strategies to enhance CPR implementation rates within the general public.

## 2. Methods

### 2.1. Study design and participants

This qualitative study, guided by the TPB, employed semi-structured interviews for data collection conducted between January 1, 2022, and May 20, 2022 [22]. To ensure credibility and rigor, the study adhered to the Consolidated Criteria for Reporting Qualitative Research (COREQ) checklist [23].

A purposive sampling strategy, utilizing snowball sampling through personal networks, was employed to recruit participants, with the aim of maximizing diversity in terms of gender, age, occupation, and education level. The sample size was determined based on the principle of data saturation, reached when interview responses became repetitive and no new themes emerged during data analysis. Consequently, 29 interviewees were included in this study [24]. For the purposes of this study, a lay rescuer was defined as an individual without specialized healthcare training or one whose training was limited to basic life support or the use of a portable defibrillator. Participants were included if they demonstrated normal communication (spoken Mandarin) and comprehension skills. Exclusion criteria encompassed interruptions in the interview process exceeding 10 min, whether due to personal matters, emotional distress, or internet connectivity issues that impeded the interview's continuation. Moreover, individuals with serious physical or psychological conditions that impaired their ability to perform CPR were excluded from the study. The demographic information of the participants is presented in Table 1, S1 Fig.

### 2.2. Interviews

This study was guided by the TPB. A comprehensive interview guide was developed through an extensive review of existing literature and consultation with experts. Following the initial interviews with the first two lay rescuers, the interview outline was revised and refined through team discussions. The finalized interview outline, depicted in Fig 2, was utilized to structure the subsequent interviews.

A total of 29 respondents from diverse age groups, occupations, and regions were recruited through both online and offline purposive sampling methods. Informed consent was obtained from all participants, either in written or electronic form. For minor participants, two-person written consent was acquired, involving both the participant and their guardian, following a phone conversation and an online consent form. Interviews were conducted either in person or online via video, with all interviewers undergoing systematic training. Two researchers were present during each interview, one conducting the interview and the other observing nonverbal cues. Prior to the interview, participants were briefed on the process, assured of its comfort, and given the option to pause or terminate at any time. Upon completion of the interview, participants received a small token of appreciation worth 10 RMB. Each interview lasted approximately 20–30 min.

### 2.3. Data analysis

The audio data were compiled and summarized within 24 h of each interview. Data analysis was conducted by a team of three researchers: A nursing professor ($Z_1$) and two postgraduate nursing students ($Z_2$ and L) trained in systematic

**Table 1. Sociology of participant population.**

| N | City Code | Gender | Age groups | Occupation | Educ | TE | WE |
|---|---|---|---|---|---|---|---|
| 1 | CS | Male | MA | Teacher (Legal Profession) | P (PhD) | Y | N |
| 2 | YH | Male | YA | Student (Economics) | U | N | N |
| 3 | KM | Female | YA | Student (Landscape Architecture) | P | Y | N |
| 4 | CZ | Female | YA | Police | U | Y | N |
| 5 | SY | Female | MA | Teacher | U | Y | N |
| 6 | CS | Male | YA | Student (Finance) | P | N | N |
| 7 | WH | Male | MA | Student (Design) | U | Y | N |
| 8 | CS | Male | MA | Businessmen | U | Y | N |
| 9 | ZZ | Male | YA | Student (Information and Computers) | U | N | N |
| 10 | CS | Male | YA | Farmer | J | N | N |
| 11 | HW | Male | YA | Student (Engineering Costs) | J | N | N |
| 12 | GZ | Female | YA | Student (International Business) | P | N | N |
| 13 | XM | Male | MA | Teacher | P | Y | N |
| 14 | CS | Male | MA | Engineers | U | N | N |
| 15 | CS | Male | YA | Student (Applied Psychology) | J | N | N |
| 16 | LD | Female | YA | Self/Media | U | N | N |
| 17 | NN | Female | YA | Student (Pharmacy) | U | Y | N |
| 18 | CZ | Female | MA | White collar (Secretary) | U | N | N |
| 19 | CS | Female | YA | Student (Architectural History and Theory) | P | N | N |
| 20 | GL | Male | YA | Student (Musicology) | U | N | N |
| 21 | KM | Female | YA | Student (Management) | P | N | N |
| 22 | CZ | Female | MA | Teacher | U | Y | N |
| 23 | XT | Female | MA | Student (law) | P | N | N |
| 24 | XA | Male | YA | Student (Electronic Information) | P | N | N |
| 25 | CZ | Female | MA | Service workers | J | N | N |
| 26 | CS | Female | MA | Civil Servants | J | N | N |
| 27 | CS | Male | MA | Freelance (Chronic Patient) | J | N | Y |
| 28 | CS | Female | MA | Unemployed | U | N | N |
| 29 | CS | Male | YA | Student (Psychology) | U | N | N |

U, undergraduate; P, postgraduate; J, junior college; YA, young adults (16~25); MA, middle-aged adults (26~45); Educ, education; TE, training experience; WE, witness experience.

qualitative research methods. Researcher $Z_2$ transcribed the audio recordings into text, which were subsequently reviewed and modified by both $Z_2$ and L to minimize information loss during the transcription process. The final transcribed texts, together with the original audio files, were imported into Nvivo 10.0, a qualitative research software, for analysis and coding. To maintain data integrity, all raw data were saved and documented for future reference. Initial coding was independently conducted by two researchers using thematic analysis [25], followed by a comparison and discussion of the identified themes. A third researcher subsequently analyzed any controversial aspects and derived conclusions directly from the raw audio recordings.

## 2.4. Credibility and rigor

To enhance the study's credibility and rigor, COREQ guidelines [23] were followed. Interviews were conducted in quiet, participant-preferred settings to promote a relaxed and comfortable environment. To establish rapport and foster trust,

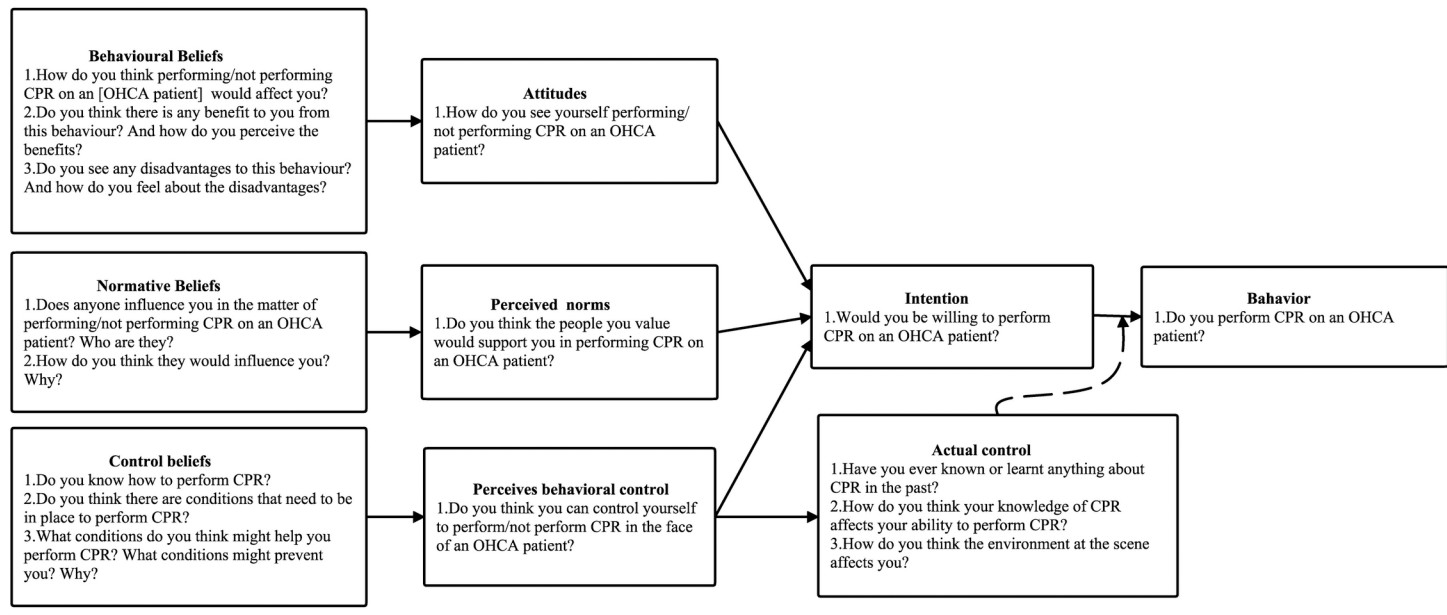

**Fig 2. Interview outline based on the TPB.**

researchers contacted participants prior to the interviews. During the sessions, researchers actively listened, paraphrased key points, and observed nonverbal cues to gain deeper insights. To ensure data quality, researchers employed a consistent interview guide and recording equipment while engaging in ongoing data analysis throughout the study. Upon completing the analysis, three participants were randomly selected for member checking, during which researchers shared the findings and sought feedback to validate the interpretations. Furthermore, reflective journaling was employed to document the researchers' coding and analysis processes, as well as the influence of their conceptual perspectives on decision-making, thereby enhancing the study's trustworthiness.

## 2.5. Ethical consideration

Informed consent was obtained from all interviewees prior to their participation in this study. To ensure participant confidentiality, all identifying information was anonymized. This study was approved by the Ethics Committee of Wuhan University People's Hospital (approval number: WDRY2021-K054).

## 3. Results

According to the TPB, the factors influencing the public's intention to perform CPR and the subsequent translation of intention into behavior encompass 11 primary themes and 6 sub-themes, as depicted in Fig 3.

### 3.1. Attitude

#### 3.1.1. Theme 1: Emotional resistance to the event of death.
CPR is intrinsically associated with death, a concept that most lay rescuers seldom encounter. Even professional rescuers often experience strong negative emotions when confronted with or even discussing death, such as feelings of guilt over failed rescue attempts and fear of mortality.

> Guilt: "Because you did not even do something correctly that caused him to die faster. This will give you a very strong sense of guilt". (P13)

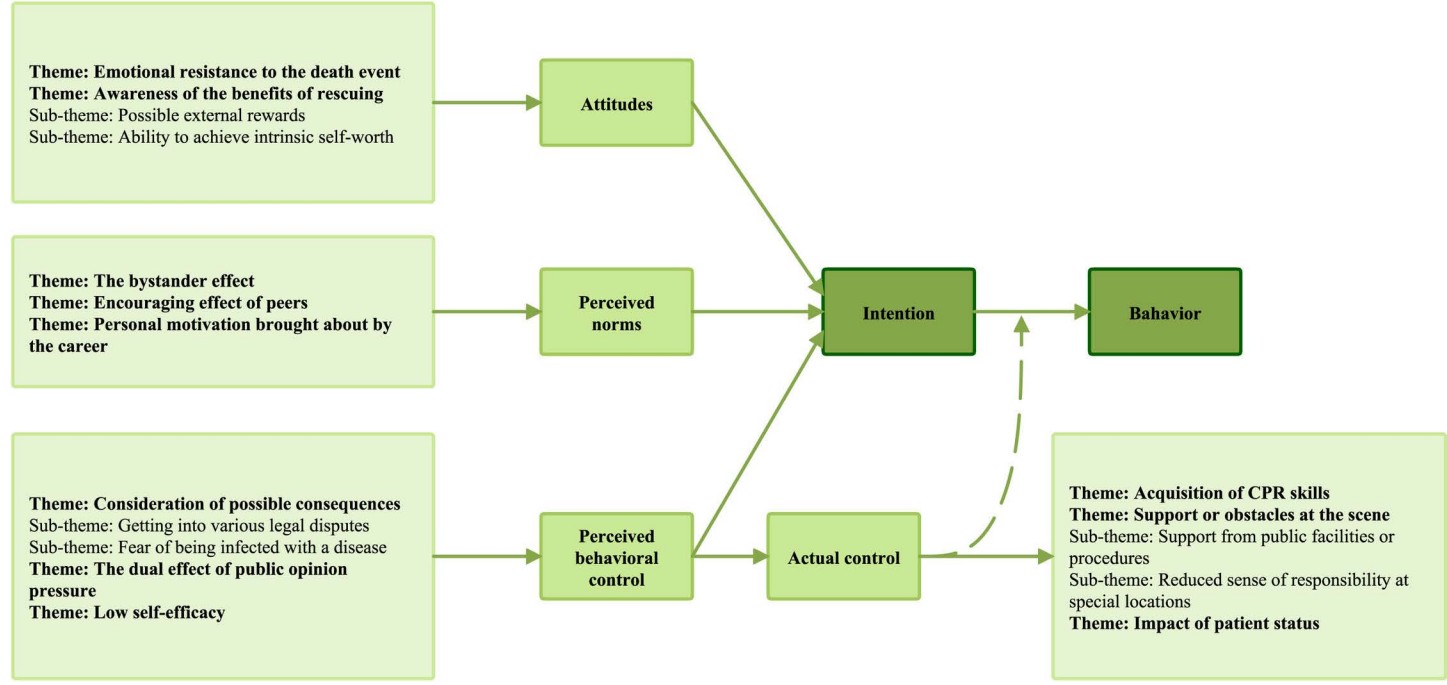

**Fig 3. Factors affecting lay rescuers' intention to perform CPR.**

*Fear: " There is a high probability that I would be willing to take action, but if the scene is too bloody or the environment is too hostile, I probably won't". (P09)*

*Fear: " It's just that I'm really afraid of death". (P27)*

### 3.1.2. Theme 2: Awareness of rewards by performing CPR.

**3.1.2.1. Sub-theme 2−1: Possible external rewards:** Lay rescuers may be motivated to perform CPR by the possibility of material or reputational rewards, such as recognition from the patient's family or community, as suggested by some participants. Moreover, personal factors, such as the patient's appearance, may influence a rescuer's willingness to intervene.

*"Well, if the patient is good-looking, then I'd accept to get involved in the resuscitation, and if the richest man in China falls to the ground, I'd also be very inclined to go up there and resuscitate him. I think I'd be paid a lot of money in return afterwards." (P09)*

**3.1.2.2. Sub-theme 2-2: Fulfillment of one's intrinsic self-worth:** Different cultural backgrounds exert a significant impact on individuals, including their willingness to help others. Traditional Chinese culture, for instance, fosters a strong sense of moral obligation to save lives, considering it a noble act. Moreover, successfully rescuing a patient can provide a profound sense of accomplishment and personal fulfillment.

*Sense of accomplishment: "Because I feel righteous. If I save someone, I also feel accomplished because I am still a young person with a sense of social justice" (P09)*

*"Saving lives is a traditional virtue of the Chinese nation! As the saying goes, saving a life is better than building a seven-layer pagoda!" (P02)*

## 3.2 Perceived norms

### 3.2.1 Theme 3: The bystander effect.

When an OHCA occurs in public settings, the presence of multiple bystanders can lead to the bystander effect, a phenomenon in which individuals experience a diminished sense of personal responsibility to intervene. Although some interviewees reported a willingness to act independently, the presence of others can contribute to a diffusion of responsibility, leading bystanders to expect that someone else will take action.

*"There were so many people around there. I thought it wouldn't be necessary for me to go, right?" (P23)*

*"I would definitely go to take a look... The noise from the people around, um, can make me even more panicked... and I might not be willing to." (P29)*

### 3.2.2. Theme 4: The encouragement from peers.

Individuals encountering an OHCA event often experience fear and a sense of overwhelm. In such situations, they may seek psychological support from nearby individuals. Peer encouragement can effectively alleviate negative emotions, motivating individuals to attempt CPR.

*"I think being with a familiar friend will give me more moral support. At least there is someone who can (stay with me). Then I can be less scared and have less worries." (P24)*

### 3.2.3. Theme 5: Motivation through personal careers.

Occupational roles exert a significant influence on individuals, frequently shaping shared personality traits among professionals. Certain professions, such as policing and teaching, tend to foster a strong sense of justice and social responsibility, which can motivate individuals to assist others, including victims of OHCA.

*"I will definitely rush to the rescue. It's the duty of the people's police, isn't it?" (P04)*

## 3.3. Perceived behavior control

### 3.3.1. Theme 6: Consideration of the possible consequences to be borne.

### 3.3.1.1. Sub-theme 6−1: Involvement in various legal disputes:

The public reported fear of becoming entangled in legal disputes, citing concerns such as the potential worsening of a patient's condition, the risk of malicious accusations from strangers, and the difficulty of proving their non-involvement in an incident. Besides, participants cited ambiguity in legal regulations and the perceived slow legal processes as key factors undermining their trust in the legal system, ultimately deterring them from intervening.

*"Because I've seen the news before. During resuscitation, the patient's ribs or something was broken, right? What if he comes after me (muttered in a low voice)." (P28)*

*"Ah, because many people have been scammed! It's very common in the community. That's why I dare not go near (patients)" (P17)*

*" But there is no detailed provision for this kind of first aid, which requires more specialization. So I don't think I have the intention to help him even if such a law is enacted."(P16)*

*"Serve your justice over such a long period of time would be physically and mentally exhausting." (P06)*

*"Therefore, even with the introduction of the relevant laws and regulations, I am still unable to provide assistance with peace of mind." (P24)*

**3.3.1.2. Sub-theme 6-2: Infection with a disease:** When introduced to the concept of CPR, respondents often associated it with artificial respiration and exposure to blood, expressing concerns about the risk of potential infection with unfamiliar diseases, particularly well-known blood-borne illnesses such as acquired immune deficiency syndrome (AIDS).

*"Because you know there's some (viruses and bacteria) in the blood, and (I'm) afraid there's some infectious disease or something. So I'd be more concerned about that, especially AIDS, right?". (P08)*

**3.3.2. Theme 7: The dual role of public opinion pressure.** The advent of technology has elevated online public opinion into a significant source of social pressure, exerting a dual influence on individuals. Some participants reported reluctance to perform CPR due to concerns about potential public condemnation in cases of poor patient prognosis. Conversely, others felt a strong compulsion to intervene, driven by fear of negative public perception for failing to assist a stranger in distress.

*"If there is no successful resuscitation, or I didn't perform CPR or something else in the right way, then it would lead to a series of negative effects on the patient's breathing. Public opinion on this matter remains strong in this community" (P07)*

*"If I wasn't there to help and it has been posted on the Internet, I would have been cursed to death" (P12)*

**3.3.3. Theme 8: Low self-efficacy.** Self-efficacy, a crucial determinant of CPR performance among lay rescuers, significantly impacts an individual's willingness to intervene during a cardiac arrest. A high level of self-efficacy can facilitate CPR implementation. However, misperceptions about CPR and unfamiliarity with CPR skills may result in diminished self-efficacy.

*"I'll call 120, but I won't do anything else. I don't think I could do it. I heard on the news that it (CPR) takes a lot of strength, and I don't think I could do it because I'm not that strong." (P25)*

*"I'm afraid to press the patient's chest because I don't use the skills often. So I don't know exactly how much to press it. I don't know or I'll have to... practice a bit before I dare to save someone's life" (P11)*

## 3.4. Actual control

**3.4.1. Theme 9: CPR skill acquisition.** CPR skill acquisition is essential for lay rescuers to translate positive intentions into concrete actions. Although participants expressed a strong desire to help, they also acknowledged feelings of helplessness due to their lack of CPR knowledge and skills. This gap in knowledge and skills ultimately hindered their ability to perform CPR.

*"I don't know CPR. If they (CPR training organization) can test me and I pass, I'll go help." (P21)*

*"I'm willing to help, even though I haven't formally learned CPR. I've witnessed the procedures before and have a basic understanding." (P27)*

**3.4.2. Theme 10: Impediments/support in the field environment.**

**3.4.2.1. Sub-theme 10−1: Support for public facilities or processes:** Reliable, well-established public facilities and proven support systems can effectively mitigate practical barriers individuals face when responding to an OHCA. The deployment of defibrillators is a crucial step in improving the prognosis of patients with cardiac arrest, while telephone-guided CPR offers a solution to address personal CPR skill deficiencies.

*"Because if you do this on your own..., the failure rate is probably higher than 90 percent. But if a professional instructs you (CPR instruction via phone), the success rate is still generally a little bit higher." (P04)*

*"Because this portable defibrillator is not as specialized and technically demanding as CPR skills. It is just relatively simple to operate." (P28)*

### 3.4.2.2. Sub-theme 10-2: Reduced responsibility at special locations:

The location of an OHCA can substantially influence a lay rescuer's decision to perform CPR. Factors such as the distance between the rescuer and the patient, proximity to a hospital, and the availability of other potential helpers can reduce a bystander's perceived sense of responsibility. Additionally, the complexity or perceived danger of the scene, particularly in cases of traumatic OHCA, may further discourage lay rescuers from intervening [26].

*"I think it depends on whether he's close to me, right? If he's far from me, then maybe someone will already be helping him before I get there." (P12)*

*"If the ambulance doesn't come, or if it's a long way (for an ambulance to the scene), I might continue to do a little bit (of rescuing). But if it's not a very long way, I'd probably choose not to do it." (P29)*

*"Ah, I probably wouldn't. I wouldn't. But maybe I would, because now, if I was at an underground station, I'd probably call the underground staff first." (P13)*

*"If it's a very, very bloody scene, I think I might be shocked at first, especially if it's a car crash." (P08)*

### 3.4.3. Theme 11: Impact of patient status.

When the patient was a child or young adult, bystanders were generally encouraged to perform CPR. However, the status of pregnant women and the elderly often discouraged bystander intervention. Interestingly, the relationship between the bystander and the patient led to contrasting responses. While some respondents expressed a willingness to perform CPR on acquaintances, others reported increased nervousness and a preference for seeking professional assistance in such situations.

*"If the patient is a child, then I definitely have to rescue ah. But pregnant women? I would not dare. I do not know if I can press (her chest)... Will it hurt the foetus in her womb ah?" (P07)*

*"If the patient is someone I know, I would do CPR. If he or she is unknown to me, I should..., or..., be afraid to do these things." (P01)*

*"I think the bystander would feel more guilty if CPR had been given to that acquaintance but with serious harm" (P21)*

## 4. Discussion

While the TPB has been extensively utilized to predict CPR intentions, its limited consideration of emotional, cultural, and contextual dynamics in real-world emergencies highlights the need for theoretical refinement. This study advances the TPB by integrating these previously underexplored dimensions, offering a more nuanced framework to account for intention-behavior gaps in bystander CPR [27–30]. It addresses three key research gaps. First, the study demonstrates how emotional barriers, such as guilt and fear associated with death, can disrupt rational decision-making processes. This emotional dimension has been underexplored within the TPB literature. Second, it highlights contextual factors, including bystander effects, legal concerns, infection risks, and patient characteristics that dynamically influence PBC. Third, the study demonstrates that despite high intentions to perform CPR, bystanders often fail to translate these intentions into actual behavior. This gap occurs due to low self-efficacy and insufficient environmental support. By integrating emotional,

cultural, and situational factors into the TPB framework, this study deepens theoretical understanding of the gap between intentions and behaviors. Besides, it offers valuable insights for developing culturally adaptive CPR training programs. These findings underscore the necessity of augmenting cognitive-behavioral models with emotional and contextual factors derived from real-world emergency responses.

## 4.1. Integrating emotional and cultural factors into the TPB attitude dimension

Our study underscores the critical need to integrate emotional responses and cultural influences into CPR training frameworks, thereby advancing a culturally informed redefinition of the "attitude" component within the TPB. Traditional CPR programs, while effective in teaching technical skills, systematically neglect emotional barriers such as guilt associated with unsuccessful resuscitation attempts or fear of confronting mortality [31]. These affective reactions—evidenced by participants' self-reports of guilt, such as concerns about "causing someone to die faster" and death-related anxiety—can significantly disrupt rational decision-making in emergency contexts [32]. This underscores the necessity of training models that integrate both cognitive assessments and emotional dynamics to improve real-world responsiveness.

The imperative to contextualize training extends to cultural dimensions. Cultural values, particularly moral obligations and the intrinsic fulfillment derived from aiding others, profoundly shape individuals' willingness to perform CPR. These emotions are not isolated psychological reactions, they are deeply shaped by cultural norms. For instance, in Confucian-influenced societies, deeply ingrained death-related taboos exacerbate emotional barriers, as individuals may fear violating cultural norms when interacting with a dying body—a concern poorly addressed by Western-centric TPB frameworks [33]. By aligning training content with culturally resonant motivators—such as framing CPR as a Confucian moral obligation in East Asian contexts or emphasizing communal responsibility in collectivist cultures— programs can amplify lay rescuers' perceived rewards and mitigate action hesitancy [34]. Notably, while cost-benefit analyses remain part of decision-making, emotional factors such as guilt and fear frequently override purely rational calculations, especially in cultural contexts where death carries stigma or spiritual significance.

To address these gaps, our refined TPB model explicitly incorporates two interrelated mechanisms: (1) The interaction between emotion-laden evaluations (guilt and fear) and cognitive processes (PBC), and (2) the moderating influence of cultural context in shaping these emotional reactions. This dual emphasis enhances both the model's explanatory power and its cross-cultural applicability. From a practical standpoint, embedding culturally specific values—such as moral obligation in collectivist societies or personal fulfillment in individualist contexts—into CPR advocacy efforts can yield more effective interventions. For example, training programs that integrate traditional narratives emphasizing heroism or community solidarity may counteract death-related taboos, thereby reframing emotional barriers as motivational drivers.

## 4.2. Contextual risks and dynamic control: Reconstructing TPB's PBC dimension

Our findings demonstrate that macro-contextual factors, particularly legal disputes and online harassment [35], significantly influence PBC in the context of bystander CPR. In contrast to traditional static views, our results reveal that PBC dynamically fluctuates based on real-time assessments of situational risks. Legal uncertainties stemming from inconsistent interpretations of Good Samaritan laws and lengthy litigation processes undermine rescuers' confidence [36]. Meanwhile, threats of online blame heighten psychological barriers by associating personal failure with public humiliation [37]. These findings highlight two key mechanisms: (1) Legal-institutional friction, where perceived control is redefined as a calculation of legal risks, and (2) sociotechnical amplification, through which digital platforms magnify reputational risks.

Conversely, the availability of external resources positively moderates both perceptions and behaviors. The presence of automated external defibrillators (AEDs) and telephone-guided CPR (T-CPR) reduces reliance on individual skills, shifting dependency toward systematic external support. AEDs enable even untrained rescuers to confidently perform lifesaving interventions by simplifying the necessary technical skills [38]. Similarly, T-CPR provides immediate professional guidance, significantly reducing uncertainty and bridging perceived skill gaps during emergency situations [39,40]. These

resources not only enhance rescuers' self-efficacy but also integrate individual actions into broader, coordinated emergency response systems. Accordingly, the availability and effective use of these external resources redefine PBC as a dynamic construct shaped through continuous interactions between individuals and support systems.

Importantly, our contextualized model of PBC explicitly integrates micro-level personal agency with macro-level structural enablers, thereby addressing criticisms of the individualistic nature of TPB. This integrated approach effectively mitigates barriers by clearly defining legal responsibilities, simplifying complex procedures, enhancing public awareness, improving legal assistance, and leveraging intelligent big-data strategies [41,42]. Moreover, systematic promotion and deployment of AEDs and T-CPR further strengthen PBC.

### 4.3. Normative conflicts and multistage intention enactment: Reconstructing TPB's subjective norm and intention-behavior pathways

Our findings underscore the inherent normative conflicts within the bystander effect, arising from the tension between competing descriptive and injunctive norms. Specifically, descriptive norms, demonstrated by others' inaction, directly contradict injunctive norms that dictate societal expectations for intervention. This conflict reveals the limitation of the TPB, which assumes normative homogeneity, thereby emphasizing the necessity for a dual-norm framework. Clearly distinguishing between these two types of norms allows for a more precise analysis of how normative conflicts affect bystander decision-making and intentions to perform CPR [43]. From a practical perspective, educational interventions should explicitly convey societal expectations, demonstrate positive bystander behaviors, and strategically leverage social pressures (roles, rewards, or punishments) to reconcile conflicting norms and facilitate CPR performance [44].

However, strong intentions alone are insufficient; practical barriers such as inadequate CPR skills, situational ambiguity, and concerns regarding the patient's condition often hinder action. These findings suggest that the TPB does not adequately account for the complexities involved in translating intentions into actual bystander rescue behaviors. Therefore, we propose a revised two-stage TPB model that explicitly distinguishes between the processes of intention formation and intention enactment [45]. This clear separation enables the development of targeted interventions: Norm-alignment strategies address normative conflicts during the intention formation stage, while ongoing CPR skills training mitigates practical barriers by improving self-efficacy during intention enactment [46,47]. Accessible CPR training workshops, real-time guidance tools (smartphone apps and hotlines), and targeted public education on managing a range of patient scenarios can significantly alleviate practical barriers.

By separating normative influences from intention-behavior stages, this refined framework effectively identifies precise intervention points, thereby significantly improving the practical outcomes of bystander CPR interventions.

### 4.4. Practical implications: Translating theory into interventions

We propose a dual-path strategy to improve bystander CPR effectiveness by combining emotionally adaptive training with systemic empowerment. This strategy addresses both emotional barriers and contextual challenges, transforming bystanders from hesitant onlookers into confident responders.

The first key component of this strategy is virtual reality (VR)-based desensitization. By simulating OHCA scenarios in a virtual environment, trainees can practice CPR in a controlled, low-risk, and high-stress setting. This immersive experience helps to mitigate emotional barriers, such as death anxiety and guilt, that often impede real-life performance. Research has consistently demonstrated that VR technology not only enhances trainees' willingness to perform CPR but also improves reaction times and self-efficacy [48]. Furthermore, it plays a critical role in reducing anxiety and fostering an overall enhancement of psychological resilience in rescuers [49].

Second, systemic empowerment strategies aim to simplify legal protections and improve access to resources, thereby further encouraging bystander involvement. One such strategy is the standardization of Good Samaritan Laws, which eliminates legal uncertainties and encourages more individuals to intervene during emergencies. Furthermore,

                                                      

the deployment of voice-guided AEDs and the provision of online maps of AED locations can lower the skill barrier for untrained bystanders, enabling more individuals to assist effectively. Empirical research has demonstrated that investments in CPR training and AED deployment significantly enhance survival rates. For instance, with a budget of S$1M, investments in CPR training and AEDs resulted in survival rates of 4.03% and 4.44%, respectively [50]. Moreover, platforms such as TikTok and WeChat promote prosocial behaviors by incentivizing participation through gamified rewards, further motivating bystanders to act during emergencies [51].

By aligning emotional readiness with systemic safeguards, our dual-path strategy bridges the intention-behavior gap, offering a culturally adaptable framework that significantly enhances the likelihood of bystander CPR in emergencies. This approach improves emotional preparedness, alleviates external pressures, and optimizes resources and legal protections, thereby contributing to a more responsive and globally effective emergency response system.

## 5. Conclusion

This study advances the TPB in the context of bystander CPR with three key contributions: (1) Incorporating emotional and cognitive factors to enhance understanding of attitude formation under stress, (2) redefining PBC as context-dependent, influenced by factors such as legal risks and AED availability, and (3) resolving conflicts between the bystander effect and moral duty through normative adjustments. To enhance intervention rates, strategies need to address both emotional and systemic barriers, such as implementing VR-based training to alleviate death anxiety, enhancing legal awareness, and expanding access to AEDs. Simplifying Good Samaritan laws can reframe PBC as a system-supported responsibility, while artificial intelligence-driven tools can influence positive social norms. Future research needs to concentrate on evaluating the cross-cultural applicability of this model and assessing interventions such as community-based AED programs and VR CPR training, with the aim of bridging the gap between hesitation and effective bystander intervention.

## 6. Limitations

Although this study offers valuable insights, several limitations warrant consideration. First, despite purposive sampling to include a diverse range of geographical and occupational backgrounds, over 60% of the participants were university-affiliated individuals (students and faculty), potentially skewing perspectives toward educated and urban populations. Second, the age distribution was predominantly weighted toward younger adults (20–30 years), thereby limiting insights into CPR decision-making among older demographics or adolescents. Third, with only one rural participant included, there is a risk of potential selection bias, which diminishes the generalizability of the findings to rural communities.

Methodologically, the reliance on online interviews due to COVID-19 restrictions may have limited researchers' ability to fully observe nonverbal cues or contextual nuances typically accessible in face-to-face interactions. Besides, the lack of CPR training among a subset of participants (exact proportion unspecified) may confound the interpretations of their self-efficacy and behavioral intentions. The study's regional focus on south-central China further restricts the applicability of its findings to culturally and socioeconomically distinct regions (western/northern China).

Lastly, the qualitative design of the study inherently prioritizes depth over breadth, and findings may not fully capture population-wide trends. While proposed future directions (such as longitudinal studies and expanded sampling) are appropriate, explicitly addressing the lack of quantitative validation (correlating themes with behavioral outcomes) and researcher bias in thematic analysis (coding subjectivity) would strengthen the critique.

## Supporting information

**S1 Fig. Participant gender and age Distribution.**
(TIF)

## Acknowledgments

We thank Home for Researchers editorial team (www.home-for-researchers.com) for language editing service.

## Author contributions

**Conceptualization:** Zhikang Zhong.

**Data curation:** Liang Liu.

**Formal analysis:** Zhikang Zhou, Weiwei Li, Lin Zhou.

**Funding acquisition:** Lin Zhou.

**Methodology:** Zhikang Zhong, Liang Liu.

**Project administration:** Liang Liu.

**Software:** Zhikang Zhong, Weiwei Lin, Lin Zhou.

**Supervision:** Liang Liu.

**Validation:** Lin Zhou.

**Writing – original draft:** Zhikang Zhong, Weiwei Lini.

**Writing – review & editing:** Liang Liu.

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
