## [Decision Letter · Decision Letter 0]

Dear Dr. LIU,

Thank you for submitting your manuscript to PLOS ONE. After careful consideration, we feel that it has merit but does not fully meet PLOS ONE’s publication criteria as it currently stands. Therefore, we invite you to submit a revised version of the manuscript that addresses the points raised during the review process.

We look forward to receiving your revised manuscript.

Kind regards,

Sebastian Schnaubelt, MD, PhD

Academic Editor

PLOS ONE

Journal Requirements:

“This study was supported by a scientific research project from the Hunan Provincial Health Commission (Project No. D202303056656), which was awarded the project to Ms ZHOU, which provided financial support for the subsequent APC.”

Reviewers' comments:

Reviewer's Responses to Questions

**Comments to the Author**

1. Is the manuscript technically sound, and do the data support the conclusions?

Reviewer #1: Yes

Reviewer #2: Partly

2. Has the statistical analysis been performed appropriately and rigorously?

Reviewer #1: Yes

Reviewer #2: N/A

3. Have the authors made all data underlying the findings in their manuscript fully available?

Reviewer #1: Yes

Reviewer #2: No

4. Is the manuscript presented in an intelligible fashion and written in standard English?

Reviewer #1: Yes

Reviewer #2: No

Reviewer #1: The is a paper discussing the factors influencing the performance of cardiopulmonary resuscitation by lay rescuers.This topic would be very important for the practical field. The paper is well-structured, with a clear introduction, methodology, results, and discussion sections. However, some sections could benefit from more detailed subheadings to guide the reader through complex arguments.

1. It would be very helpful for the authors to re-organise the literature review. Especially, please ensure that the theoretical framework is consistently referenced throughout the paper. This will help to tie the various sections together and reinforce the central arguments.

2. The existing theoretical framework is almost fully based on extended TPB. It would be important for the study to highlight the theoretical novelties by highlighting the difference between this work and other similar publications.

3. Please provide a more detailed background of data collection process and some more backgrounds of the 29 participants.

4. It would be important to highlight the implications of this research by combining the findings with practical policies and treatment. Highlight how the findings can be applied in real-world settings. This could involve discussing potential interventions, policy implications, or practical applications in various fields such as education, healthcare, or organizational behavior. Some interventions would be very helpful, for example novel interventions presented in “Pricing strategy for household energy-saving option (HESO): A novel option-based intervention for promoting household energy efficiency” would be a potential example.

5. A more detailed discussion of the study’s limitations and how they might impact the findings would provide a more balanced perspective. This could include potential biases, limitations in the data collection process, and any assumptions made.

6. While the paper is academically rigorous, consider simplifying some of the language to make it more accessible to a broader audience.

Reviewer #2: This is a very interesting topic in the background where bystander CPR rates are either declining or stagnant despite major efforts.

However, the language needs to be improved to effectively communicate what authors have done and found.

**Do you want your identity to be public for this peer review?** For information about this choice, including consent withdrawal, please see our Privacy Policy

Reviewer #1: No

Reviewer #2: **Yes: ** Fahad Javaid Siddiqui

---

## [Author Response · Author response to Decision Letter 1]

18 Dec 2024

Dear Editors:

Thank you for your kind letter and your careful work regarding our manuscript. We have revised the manuscript in accordance with your comments. And point-by-point responses to the comments were as follows:

1 Please ensure that your manuscript meets PLOS ONE's style requirements, including those for file naming.

Response: We sincerely thank the editors for their careful review of our manuscript. In response to the feedback provided, we have made the following adjustments: (a) We have reorganized the manuscript’s sections to improve its clarity and logic, ensuring its alignment with PLOS ONE’s style requirements. (b) All files have been renamed in accordance with PLOS ONE’s file naming requirements. We hope these changes addressed the revision requirements and contribute to a clearer presentation of our work. We look forward to your further feedback and hope that our manuscript is now fully aligned with the journal’s formatting expectations.

“This study was supported by a scientific research project from the Hunan Provincial Health Commission (Project No. D202303056656), which was awarded the project to Ms ZHOU Lin, which provided financial support for the subsequent APC.” (Line 15, page 26)

Response: We sincerely thank the editors for their careful review, and we apologize for any oversight on our part. Ms. Zhou Lin, as the principal investigator, provided significant contributions to the formal analysis of the data, the interpretation of results, and the revision of the manuscript. We have now included this information in the “Author Contributions” section to accurately reflect her role in the study.

Response: We appreciate your feedback regarding the data availability statement. Thank you for your feedback regarding the data availability statement. We fully support PLOS ONE's requirements concerning data availability, as they enhance the quality of research. We have revised the data availability statement in accordance with your suggestions: All additional supporting information files are accessible through the openICPSR database at the following link: https://doi.org/10.3886/E213184V2. We hope that these modifications will better align with PLOS ONE's requirements for data availability.

Dear reviewers:

Thank you for your careful review and constructive suggestions regarding our manuscript. We have revised the manuscript in accordance with the comments and marked all the emendation on our revised manuscript.

Reviewer #1: The is a paper discussing the factors influencing the performance of cardiopulmonary resuscitation by lay rescuers. This topic would be very important for the practical field. The paper is well-structured, with a clear introduction, methodology, results, and discussion sections. However, some sections could benefit from more detailed subheadings to guide the reader through complex arguments.

1. It would be very helpful for the authors to re-organise the literature review. Especially, please ensure that the theoretical framework is consistently referenced throughout the paper. This will help to tie the various sections together and reinforce the central arguments.

Response: We would like to thank the Reviewer #1 for the constructive feedback. In response, we have reorganized the manuscript to improve clarity and strengthen the connections between key concepts. The following changes were made:

In response to the reviewers' comments, we made significant revisions to the literature review, streamlined the research background, and concentrated on the influencing factors (barriers, (Line 4, page 5)) related to the exploration of bystander cardiopulmonary resuscitation through the lens of the Theory of Planned Behavior (TPB). Additionally, we incorporated visual elements to enhance the clarity of the main ideas presented. (Line 11, page 6) Moreover, the theoretical framework of TPB has been consistently applied throughout the data collection and analysis process (Line 2, page 8) Finally, in the discussion section, we focused on the interplay between the TPB components: “Attitude”, “Subjective Norm”, and “Perceived Behavioral Control”. We restructured the discussion into three key themes based on practical considerations: Adaptation of CPR Training Strategies (Line 12, page 18), Updates to Laws and Policies (Line 4, page 20) and Control of Real-World Environments (Line 22 page 22)

This structure aims to provide clearer and more actionable insights that are directly tied to the TPB framework and its practical implications.

In summary, these revisions are intended to unify the central theory of the article and strengthen the connections between the theoretical basis and the practical recommendations. By focusing on the interaction between TPB components and dividing the discussion into actionable themes, we believe the manuscript offers a clearer and more cohesive direction for future interventions.

2. The existing theoretical framework is almost fully based on extended TPB. It would be important for the study to highlight the theoretical novelties by highlighting the difference between this work and other similar publications.

Response: We sincerely thank the Reviewer #1 for the constructive comments on the article. In response to the feedback, we have revisited the existing literature and emphasized the theoretical novelties of our work. As stated in the manuscript (Line 14, page 5), “It is important to note that there are currently limited studies that examine the experiences of lay rescuers perform CPR within the framework of this theory.” Most existing studies in this field have employed cross-sectional or quantitative methodologies, focusing on the validity of the TPB model in various contexts. For example:

A qualitative study by Andrews et al. (2018) explored factors influencing healthcare professionals’ use of Automated External Defibrillators (AEDs) in hospital settings, offering valuable insights into healthcare providers’ decision-making processes [Andrews et al., 2018].

Panchal et al. (2015) proposed an “Intention-Focused” paradigm for improving bystander CPR performance, but this work remains relatively broad and lacks an in-depth qualitative exploration of bystander experiences [Panchal et al., 2015].

In contrast, our study is among the few contemporary qualitative investigations that specifically examine the experiences of lay rescuers performing bystander CPR within the TPB framework. Our research bridges an important gap by exploring how TPB components—such as attitude, subjective norms, and perceived behavioral control—interact in real-world CPR situations. Furthermore, our study highlights the lived experiences of laypersons, which contrasts with prior work that has focused on more formal or institutional settings (e.g., healthcare workers). We believe that this experiential perspective offers valuable insights into the barriers and facilitators of CPR behavior among non-experts, and provides a nuanced understanding of how to design more effective interventions aimed at encouraging CPR in the general population.

3. Please provide a more detailed background of data collection process and some more backgrounds of the 29 participants.

Response: We would like to express our gratitude to the Reviewer #1 for the thorough and insightful comments regarding this section. In response, we have enhanced the details of the entire study process within the “Study Design & Participants” sections. ( Line 13, page 6) “Interviews” ( Line 2, page 8) and “Credibility and Rigor”. ( Line 18, page 9) Furthermore, we have included additional demographic information about the participants, specifically regarding their prior participation in CPR training and any experiences they may have had with CPR rescue. We believe that these enhancements will further strengthen the rigor of the data.

4. It would be important to highlight the implications of this research by combining the findings with practical policies and treatment. Highlight how the findings can be applied in real-world settings. This could involve discussing potential interventions, policy implications, or practical applications in various fields such as education, healthcare, or organizational behavior. Some interventions would be very helpful, for example novel interventions presented in “Pricing strategy for household energy-saving option (HESO): A novel option-based intervention for promoting household energy efficiency” would be a potential example.

Response: Thank you for your insightful suggestion to emphasize the practical implications of our research. We have incorporated a more detailed discussion of how our findings can be applied in real-world settings. In the revised article, we have incorporated interventions that bridge research findings with practical application. When adjusting the CPR training strategy, we introduced the concept that first aid training should be integrated with death education. (Line 3, page 19) Furthermore, it is important to recognize that current first aid training primarily emphasizes quantity over quality, indicating that additional resources are necessary to address this issue (Line 21, page 19). Regarding the updating of laws and policies, based on the research findings, we advocate for guiding the public to foster an 'altruistic' mentality while also respecting the 'self-benefiting' mentality. We suggest implementing incentive measures to enhance public enthusiasm for rescue efforts (Line 17, page 20). Additionally, in light of the complexities of legal frameworks, establishing legal foundations and strengthening legal assistance may more effectively resolve the public's legal dilemmas. (Line 16, page 21) In response to the emergence of 'we media,' utilizing invisible tools based on AI and big data models to guide public opinion is a crucial strategy to address the interests and challenges posed by the Internet. (Line 14, page 22) When managing the physical environment, it is essential to acknowledge its complexity; thus, creating a more accessible first aid environment—such as telephone-based first aid and easy access to defibrillators—can significantly enhance the effectiveness of first aid interventions (Line 16, page 23). Moreover, given the public's concerns regarding the first aid environment, further optimization of this environment should remain a priority moving forward. In summary, by implementing these strategies, the research findings can contribute to improving lay rescuers' effectiveness in real-world scenarios, ultimately leading to better health outcomes and increased survival rates from cardiac arrests. We hope this expanded discussion effectively addresses your recommendation and provides a clearer connection between the research findings and their practical applications.

5. A more detailed discussion of the study’s limitations and how they might impact the findings would provide a more balanced perspective. This could include potential biases, limitations in the data collection process, and any assumptions made.

Response: Thank you for highlighting the need to include a more detailed discussion of the study’s limitations. In response, we have expanded this section to provide a more balanced perspective, addressing potential biases, data collection challenges, and underlying assumptions. Specifically, the revised section now discusses the representativeness of the sample, including factors such as campus background, age, and geographical area. Furthermore, it examines the influence of cultural and environmental contexts, as well as the methods of data collection—such as the use of partially online and single semi-structured interviews—and the process through which research findings were derived (Line 12, page 25).We believe this expanded discussion not only offers a more nuanced perspective on the study’s findings but also enhances the overall rigor and transparency of the research.

6. While the paper is academically rigorous, consider simplifying some of the language to make it more accessible to a broader audience.

Response: We sincerely thank the Reviewer #1 for the recognition of this study. Our team fully agrees with your perspective. To improve clarity and accessibility, we have simplified the language throughout the article and incorporated visual elements in certain sections to better illustrate key concepts. Additionally, we enlisted the assistance of native English speakers to enhance the clarity of the paper, ensuring it aligns with the journal's writing standards.

7.Reviewer #2: This is a very interesting topic in the background where bystander CPR rates are either declining or stagnant despite major efforts. However, the language needs to be improved to effectively communicate what authors have done and found.

Response: We sincerely thank the Reviewer #2 for the positive feedback and affirmation of this study. We greatly value your inquiries and, as a result, have thoroughly reviewed the entire text and made necessary revisions. Additionally, we sought the assistance of native English speakers to further refine the language, ensuring that the core ideas of the article are conveyed more effectively.

---

## [Decision Letter · Decision Letter 1]

Dear Dr. LIU,

Thank you for submitting your manuscript to PLOS ONE. After careful consideration, we feel that it has merit but does not fully meet PLOS ONE’s publication criteria as it currently stands. Therefore, we invite you to submit a revised version of the manuscript that addresses the points raised during the review process.

Thank you!

We look forward to receiving your revised manuscript.

Kind regards,

Sebastian Schnaubelt, MD, PhD

Academic Editor

PLOS ONE

Additional Editor Comments :

Dear authors,

I have invited another reviewer due to conflicting reports in the previous round. Please address the new reviewer's comments, and try again revising your manuscript according to reviewer 1's comments.

Thank you!

Reviewers' comments:

Reviewer's Responses to Questions

**Comments to the Author**

Reviewer #1: (No Response)

Reviewer #2: All comments have been addressed

Reviewer #3: All comments have been addressed

2. Is the manuscript technically sound, and do the data support the conclusions?

Reviewer #1: No

Reviewer #2: Yes

Reviewer #3: Yes

3. Has the statistical analysis been performed appropriately and rigorously?

Reviewer #1: No

Reviewer #2: N/A

Reviewer #3: Yes

4. Have the authors made all data underlying the findings in their manuscript fully available?

Reviewer #1: No

Reviewer #2: No

Reviewer #3: Yes

5. Is the manuscript presented in an intelligible fashion and written in standard English?

Reviewer #1: No

Reviewer #2: Yes

Reviewer #3: Yes

Reviewer #1: The is a very interesting study. However, the study exist some limitations that should be addressed. (1) The theory employed in this study has been widely used, especially in the relevant topics. The manuscript failed to prove its contribution to the theory and the knowledge body. (2)It is believed that the sample is not sufficient enough to support the findings and statements and the revised version also dose not provide sufficient discussion and evidence for this.

Reviewer #2: The langauge and structural organization improvement has made the paper a contribution towards this topic.

Reviewer #3: Thank you for an interesting paper on cultural insights of China. Thank you for providing reviewers’ comments and responses to them – interesting to track the progress of the project.

Major points

Methods

Page 6, Line 18. How were the participants recruited? Social media, personal connections, some organizations/networks?

Page 7, Line 2. What do you mean by ‘normal communication and comprehension skills’? Has everyone spoken Mandarin? Was the interpreter involved if otherwise?

Page 7, Line 3. What do you mean by ‘interruptions’? Any interruptions? For how long?

Table 1. Why did you use age ranges instead of actual values? Some meaningful age categories, such as ‘young adults’, etc, with percentages would describe the population better. The age range is 16-45, with only one minor and one in the range of 40-45 – doesn’t look that diverse in terms of age (Page 8, Line 8).

Participant 18 (White Collar) and Participant 27 (Freelance) – what are actual specialties of these people?

Section 2.2. Interviews. Have you developed your own interview or adapted one based on literature review. If yes, how did you adapt and translate it from Chinese to English and vice versa? How was the quality check performed to make sure that the wording in Chinese is exactly how you intended to sound and does not prompt the respondents?

Page 8, Line 6. How was the interview revised and refined?

Page 8, Line 10-11. I found only one minor in the table (Participant 10). What about the second minor participant?

Page 9, Line 19. Could you submit the filled COREQ checklist?

Discussion

Page 20, Line 16. Propaganda can be felt on social media. Algorhythms might also prevent the general population without specific interest in watching such content.

Page 24, Lines 13-14. What to do with pregnant and elderly people then?

Minor points

Introduction

Page 4, Line 5. Please, cite AHA Guidelines.

Results

Page 11, Line 4. Sometimes you mention (P13), and sometimes just put numbers. Please, be consistent – it might be a bit confusing. First I thought it was a citation.

Please, be more cautious about the interpretation of results. Action might be different from concerns and worries, especially without actual experience.

**Do you want your identity to be public for this peer review?** For information about this choice, including consent withdrawal, please see our Privacy Policy

Reviewer #1: No

Reviewer #2: No

Reviewer #3: No

---

## [Author Response · Author response to Decision Letter 2]

12 Jun 2025

Dear Editors

Thank you for your thoughtful letter and the thorough review of our manuscript. We have revised the manuscript in accordance with the reviewers' comments. Below are our point-by-point responses to each of the comments:

Reviewer #1-1: The theory employed in this study has been widely used, especially in the relevant topics. The manuscript failed to prove its contribution to the theory and the knowledge body.

Response to Reviewer’s Comment:

Thank you for highlighting this important aspect. We appreciate the opportunity to clarify how our study advances the Theory of Planned Behavior (TPB) and contributes to the broader body of knowledge in bystander CPR research. Specifically, our manuscript offers three distinct theoretical contributions, which we have explicitly articulated in the revised Discussion section:

1. Integration of Emotional and Cultural Factors into TPB’s Attitude Dimension (page 20, line 2): Our study extends the TPB’s attitude dimension by incorporating emotional and cultural contexts, thereby offering deeper insights into the motivational factors that influence bystanders’ CPR decisions.

2. Contextual Risks and Dynamic Control: Reconstructing TPB’s Perceived Behavioral Control (PBC) Dimension (page 21, line 17): We address limitations in the traditional conceptualization of PBC by emphasizing the dynamic nature of perceived control, particularly how situational risks and contextual variables significantly affect perceived control. This enhances the predictive accuracy and practical applicability of the TPB in emergency situations.

3. Normative Conflicts and Multistage Intention Enactment: Reconstructing TPB’s Subjective Norm and Intention-Behavior Pathways (page 23, line 14): Our study identifies and addresses critical gaps related to normative conflicts, emphasizing the complex, multistage process linking intention formation and behavioral enactment. This refined perspective recognizes that normative pressures and social conflicts dynamically shape bystanders’ intentions and subsequent behaviors.

Furthermore, our study proposes a dual-path strategy to enhance bystander CPR effectiveness. By combining emotionally adaptive training with systemic empowerment, this strategy directly addresses emotional barriers and contextual challenges, transforming hesitant bystanders into confident responders (page 24, line 12).

Through the integration of cultural and contextual nuances into the TPB framework, our research not only advances theoretical development but also lays a robust foundation for designing cross-cultural CPR interventions. The revised Discussion section (pages 19–25) explicitly delineates these theoretical advancements and their broader implications for future research.

Thank you again for your valuable feedback, which helped us clarify the theoretical contributions and implications for CPR training.

Reviewer #1-2: It is believed that the sample is insufficient to support the findings and statements, and the revised version does not provide adequate discussion or evidence to address this.

Response to Reviewer’s Comment:

Thank you for highlighting the concern regarding the sample size and the sufficiency of evidence. We recognize that sample size is a critical consideration in qualitative research. Below, we clarify our methodological approach and demonstrate how the revised manuscript strengthens the discussion on limitations and generalizability.

1. Sample Adequacy in Qualitative Research:

Our study employed purposeful sampling, guided by the principle of topic saturation, which is widely accepted in qualitative research (Birks & Mills, 2015; Olshansky, 2015; Given, 2016). We closely monitored saturation, noting that no new themes emerged after the 26th participant. To confirm redundancy, we interviewed an additional three participants, bringing the total to 29. This approach aligns with the recommendations of Hennink et al. (2017), who emphasize the in-depth exploration of complex psychosocial phenomena rather than statistical quantification.

While the sample size (N=29) may seem modest, it is consistent with best practices in qualitative research, which prioritize depth of understanding over sample size. In qualitative studies focused on understanding nuanced experiences, a smaller sample is often sufficient to reach saturation and generate meaningful insights.

2. Addressing Generalizability and Limitations:

To improve transparency, we have added a "Limitations" section (page 26, line 13), where we discuss potential limitations and the generalizability of our findings:

• "Although this study provides valuable insights, several limitations should be noted. First, while purposive sampling aimed to capture a diverse range of geographical and occupational backgrounds, more than 60% of participants were university-affiliated (students and faculty), which may skew the findings toward the perspectives of educated, urban populations. Second, the sample was predominantly composed of younger adults (ages 20-30), which may limit the applicability of our findings to older populations or adolescents. Finally, with only one rural participant, there is a risk of selection bias, which may affect the generalizability of our findings to rural communities."

We trust that this revised discussion clarifies our methodological choices and addresses the study’s limitations, thereby strengthening the transparency and rigor of our research.

Reviewer #3-1�How were the participants recruited? Social media, personal connections, some organizations/networks?

Response to Reviewer’s Comment:

Thank you for raising this important question. We describe our two-fold participant recruitment approach in the methodology section (Page 7, Line 9). First, we initiated recruitment through the researchers' personal networks to establish core participants. Subsequently, a snowball sampling technique was employed, where existing participants helped identify additional qualified candidates who met our diversity criteria across gender, age, occupation, and education levels. This strategy effectively balanced accessibility with our commitment to demographic variety while maintaining ethical recruitment standards.

Reviewer #3-2�What do you mean by ‘normal communication and comprehension skills’? Has everyone spoken Mandarin? Was the interpreter involved if otherwise?

Response to Reviewer’s Comment:

Thank you for prompting us to elaborate on this critical methodological aspect. The requirement of "normal communication and comprehension skills" involved a dual validation process:

Communication competence was assessed through:

• Mandarin fluency verification: All participants completed a 5-minute unstructured conversation during recruitment screening to evaluate spontaneous speech clarity and dialect neutrality.

• Contextual responsiveness: Ability to maintain topic relevance and demonstrate reciprocal dialogue patterns during preliminary discussions.

Comprehension capacity was operationalized as:

• Consistent logical coherence in responding to multi-layered interview questions.

As explicitly stated in our inclusion criteria (Page 7, Line 17), Mandarin proficiency was a prerequisite, with the following safeguards:

• Pre-screening filter: Self-reported Mandarin as the primary daily language (>5 years of continuous use).

No interpreters were employed.

Reviewer #3-3�What do you mean by ‘interruptions’? Any interruptions? For how long?

Response to Reviewer’s Comment:

Thank you for allowing us to clarify this procedural detail. In this context, interruptions were operationally defined as unplanned discontinuities that:

Originated from participants (e.g., urgent personal obligations, emotional instability)

Persisted beyond 10 minutes despite researcher efforts to resume

Fundamentally disrupted the interview's narrative flow (assessed via post-interview audio analysis)

We implemented a tiered response protocol:

<5 min pauses: Documented but permitted continuation

5-10 min interruptions: Triggered protocol review with optional rescheduling

>10 min discontinuities: Systematically excluded per pre-defined criteria (Page 7, Line 18)

Notably, technical interruptions (e.g., internet instability) were handled through real-time troubleshooting before applying exclusion measures. All exclusion decisions underwent dual verification by two researchers to prevent arbitrary implementation.

Reviewer #3-4�Table 1. Why did you use age ranges instead of actual values? Some meaningful age categories, such as ‘young adults’, etc, with percentages would describe the population better. The age range is 16-45, with only one minor and one in the range of 40-45 – doesn’t look that diverse in terms of age.

Response to Reviewer Comments:

We sincerely thank the reviewer for the valuable comments regarding our age grouping and the diversity of the sample. In response to your concerns about the use of age ranges and the limited age diversity, we provide the following explanations and revisions:

1. Rationale for Using Age Ranges

In the initial design of our study, we adopted age ranges based on common practices in related fields to simplify data presentation and analysis. Moreover, given the limited sample size, further subdividing age groups could have led to insufficient numbers in each subgroup, thus reducing statistical power. Nonetheless, we acknowledge that more detailed age categories, including percentages, would enhance transparency and better meet readers’ expectations.

2. Acknowledgement of Limitations

As the reviewer pointed out, the current sample mainly covers the age range 16-45 years, with few participants under 18 and in the 40-45 bracket, resulting in limited age diversity. We have explicitly discussed this limitation in the revised manuscript’s “Limitations” section. Specifically, we noted that the age distribution is predominantly weighted toward younger adults (20–30 years), which restricts insights into decision-making among adolescents and older adults. This limitation reflects the challenges of recruiting a fully age-diverse sample, especially under the constraints of the COVID-19 pandemic and available recruitment channels.

3. Revisions and Improvements

To improve transparency, we have refined the age classification in the revised manuscript by adding two groups: Young Adults (YA, 16-25 years) and Middle-aged Adults (MA, 26-45 years). We have also updated the relevant table to include both the number and percentage of participants in each group to provide a clearer picture of the age distribution.

4. Explanation of Age Diversity and Future Directions

The limited age diversity observed is largely due to recruitment challenges, including reliance on online interviews and purposive sampling during the pandemic, which may have excluded older adults and minors. Future research will aim to broaden recruitment strategies to include more minors and participants over 45 years old. Additionally, expanding the geographical and demographic scope will help generalize findings across different age groups and communities. We also acknowledge that the qualitative design prioritizes depth over breadth, and future quantitative studies could validate these findings by correlating age-related themes with behavioral outcomes.

We appreciate the reviewer’s insightful comments, which have greatly contributed to enhancing the rigor and clarity of our manuscript.

Reviewer #3-5�Participant 18 (White Collar) and Participant 27 (Freelance) – what are actual specialties of these people?

Response to Reviewer Comments:

Thank you for your query. We have verified the backgrounds of the participants in question:

• Participant 18 (originally labeled "White Collar"):

Actual specialty: Running an online shop (e-commerce operations).

Clarification: While initially categorized broadly as "white-collar," his specific expertise lies in managing digital storefronts, inventory, and online sales.

• Participant 27 (originally labeled "Freelance"):

Actual specialty: Chronic hemodialysis patient.

Clarification: His "freelance" label referred to incidental work arrangements, but his core identity relevant to this study is his lived experience as a long-term dialysis patient.

Reviewer #3-6�Interviews. Have you developed your own interview or adapted one based on literature review. If yes, how did you adapt and translate it from Chinese to English and vice versa? How was the quality check performed to make sure that the wording in Chinese is exactly how you intended to sound and does not prompt the respondents? How was the interview revised and refined?

Response to Reviewer

Thank you for your valuable feedback on our interview methodology. We appreciate the opportunity to clarify our approach:

1. Interview Development and Cross-Language Adaptation

We designed a semi-structured interview guide, drawing upon both Chinese and English literature to ensure theoretical rigor and cultural relevance. The adaptation process followed these steps:

a. Bilingual Literature Synthesis:

• Conceptual Alignment: For terms or constructs with unclear or ambiguous cross-cultural equivalence, we consulted domain experts to ensure accurate conceptual representation in both languages. Additionally, bilingual scholars were engaged to resolve any translation discrepancies.

b. Translation Protocol:

• Forward-Backward Translation: A bilingual researcher translated the interview guide from Chinese to English, focusing on conceptual meaning rather than literal translation. A second independent translator performed a back-translation into Chinese. Discrepancies between the original and back-translated versions were thoroughly discussed and resolved by the research team.

• Cultural Calibration: Native speakers of both English and Mandarin reviewed the phrasing to ensure linguistic naturalness, cultural appropriateness, and neutrality.

2. Quality Assurance Mechanisms

Several strategies were employed to ensure linguistic accuracy and minimize response bias:

a. Cognitive Pretesting:

• Participant Paraphrasing: During the pilot phase, participants were asked to paraphrase interview questions to confirm clear understanding of their intent and meaning.

• Nonverbal Cue Monitoring: Interviewers monitored participants’ nonverbal cues (e.g., pauses, signs of confusion) to identify areas requiring clarification or rephrasing.

b. Neutrality Audits:

• Review of Question Language: Each question was reviewed to eliminate assumptions and prevent leading or biased language.

• Member Checking: Post-interview, participants confirmed the accuracy of their responses, ensuring the clarity and neutrality of the questions.

3. Iterative Refinement During Data Collection

The interview guide was continuously refined based on feedback obtained throughout data collection:

• Early-Interview Adjustments: Following interviews with the first 3-5 participants, any confusing or inconsistently interpreted questions were revised.

• Emergent Theme Integration: New themes identified during interviews led to adjustments in the guide, incorporating probes to explore these topics more thoroughly.

4. Validation of Final Instrument

o Thematic Saturation: Data collection continued until no new themes emerged, confirming the adequacy and comprehensiveness of the interview guide.

o Cross-Language Consistency: Final translations were tested with bilingual participants to ensure equivalence and accuracy between languages.

These methods ensured that our interview instrument was both culturally sensitive and methodologically rigorous. We also took proactive steps to avoid bias and leading questions in our design. We hope this detailed explanation addresses your concerns, and we are happy to provide further documentation if needed.

Reviewer #3-6�I found only one minor in the table (Participant 10). What about the second minor participant?

Response to Reviewer

We appreciate the reviewer’s feedback regarding the participants. After reviewing our records, we confirm that only one minor (Participant 10) was included in the study. To ensure ethical compliance and respect for the participant's decision-making capacity, informed consent was obtained from both the minor and their primary guardian.

Reviewer #3-7�Could you submit the filled C

---

## [Editor Report · Decision Letter 2]

Factors influencing the performance of cardiopulmonary resuscitation by lay rescuers: a qualitative study based on the Theory of Planned Behavior

PONE-D-24-25699R2

Dear Dr. LIU,

We’re pleased to inform you that your manuscript has been judged scientifically suitable for publication and will be formally accepted for publication once it meets all outstanding technical requirements.

Kind regards,

Sebastian Schnaubelt, MD, PhD

Academic Editor

PLOS ONE

---

## [Editor Report · Acceptance letter]

PONE-D-24-25699R2

PLOS ONE

Dear Dr. Liang,

I'm pleased to inform you that your manuscript has been deemed suitable for publication in PLOS ONE. Congratulations! Your manuscript is now being handed over to our production team.

Kind regards,

on behalf of

Dr. Sebastian Schnaubelt

Academic Editor

PLOS ONE
